# FRAGMENT-BASED SEQUENTIAL TRANSLATION FOR MOLECULAR OPTIMIZATION

## ABSTRACT

Searching for novel molecular compounds with desired properties is an important problem in drug discovery. Many existing frameworks generate molecules one atom at a time. We instead propose a flexible editing paradigm that generates molecules using learned molecular fragments—meaningful substructures of molecules. To do so, we train a variational autoencoder (VAE) to encode molecular fragments in a coherent latent space, which we then utilize as a vocabulary for editing molecules to explore the complex chemical property space. Equipped with the learned fragment vocabulary, we propose **Fr**agment-based **S**equential **T**ranslation (FaST), which learns a reinforcement learning (RL) policy to iteratively translate model-discovered molecules into increasingly novel molecules while satisfying desired properties. Empirical evaluation shows that FaST significantly improves over state-of-the-art methods on benchmark single/multi-objective molecular optimization tasks.

## 1 INTRODUCTION

Molecular optimization is a challenging task that is pivotal to drug discovery applications. Part of the challenge stems from the difficulty of exploration in the molecular space: not only are there physical constraints on molecules (molecular strings/graphs have to obey specific chemical principles), molecular property landscapes are also very complex and difficult to characterize: small changes in the molecular space can lead to large deviations in the property space.

Recent fragment-based molecular generative models have shown significant empirical advantages (Jin et al., 2019a; Podda et al., 2020; Xie et al., 2021) over atom-by-atom generative models in molecular optimization. However, they generally operate over a fixed set of fragments which limit the generative capabilities of these models. Shifting away from previous frameworks, we learn a distribution of molecular fragments using vector-quantized variational autoencoders (VQ-VAE) (van den Oord et al., 2017). Our method builds molecular graphs through the addition and deletion of molecular fragments from the learned distributional fragment vocabulary, enabling the generative model to span a much larger chemical space than models with a fixed fragment vocabulary. Considering atomic edits as primitive actions, the idea of using fragments can be thought of as *options* (Sutton et al., 1999; Stolle & Precup, 2002) as a temporal abstraction to simplify the search problem.

We further introduce a novel *sequential translation* scheme designed for fragment-based molecular optimization. We start the molecular search by translating from known active molecules and store the discovered molecules as new potential initialization states for subsequent searches; we incorporate a delete action in our model, enabling our method to backtrack to good molecular states. Previous works optimize molecules either by generating from scratch or a single translation from known molecules, which is inefficient in finding high-quality molecules and often discovering molecules lacking novelty/diversity. Our proposed framework addresses these deficiencies since our method is (1) very efficient in finding molecules that satisfy property constraints as the model stay close to the high-property-score chemical manifold; and (2) able to produce highly novel molecules with our flexible learned fragment vocabulary and a sequence of fragment-based editing.

Combining the advantage of a distributional fragment vocabulary and the sequential translation scheme, we propose **Fr**agment-based **S**equential **T**ranslation (FaST), which is realized by an RL policy that proposes fragment addition/deletion to a given molecule. Our proposed method can generate molecules under various objectives such as property constraints, novelty constraints, and diversity constraints. The main contribution of this paper includes:

1. We demonstrate a way to learn distributional molecular fragment vocabulary through a VQ-VAE and the effectiveness of the learned vocabulary in molecular optimization.

2. We propose a novel molecular search scheme of sequential translation, which gradually improves the quality and novelty of generation through backtracking and a stored frontier.

3. We implement a RL policy combining the fragment vocabulary and the sequential translation scheme that significantly outperforms state-of-the-art methods in benchmark single/multi-objective molecular optimization tasks.

## 2 RELATED WORK

**Molecular generation and optimization.** Early works on molecular optimization build on generative models on both SMILES/SELFIES string (Gómez-Bombarelli et al., 2018; Segler et al., 2018; Kang & Cho, 2018; Krenn et al., 2020; Nigam et al., 2021b; Shin et al., 2021), and molecular graphs (Simonovsky & Komodakis, 2018; Liu et al., 2018; Ma et al., 2018; De Cao & Kipf, 2018; Samanta et al., 2020; Mercado et al., 2021) and generate molecules character-by-character or node-by-node. Jin et al. (2018) generates graphs as junction trees by considering the vocabulary as the set of atoms or predefined rings from the data; Jin et al. (2020) use the same atom+ring vocabulary to generate molecules by augmenting extracted rationales of molecules.

**Generating molecules with a fixed molecular fragment vocabulary** is a well-established idea in traditional drug design (Erlanson, 2011), and has been recently explored through deep learning models (Podda et al., 2020; Xie et al., 2021; Kong et al., 2021; Fu et al., 2021; 2020; Leguy et al., 2020), outperforming previous atom-level models. Recent synthesizability-aware models can also generate single-step reaction (Bradshaw et al., 2019) and molecule synthesis graphs (Bradshaw et al., 2020) based on a fixed reactant pool. However, the fixed fragment vocabularies used by these models, which are typically small and predefined a priori, limit the chemical space spanned by the models. In our work, we utilize a learned molecular fragment vocabulary, which is obtained by training a VQ-VAE on a large set of fragments extracted from ChEMBL (Gaulton et al., 2012). By sampling fragments from the learned distribution, our model can span a much larger chemical space than methods using a fixed vocabulary (visualized in Figure 3a, Figure 3b).

**Sequential generation of molecules.** Guimaraes et al. (2017); Olivecrona et al. (2017); You et al. (2018); Zhou et al. (2019) frame the molecular optimization problem as a reinforcement learning problem, but they generate on the atom/character level and from scratch each time, reducing the efficiency of the search algorithm. Jin et al. (2019b) uses a graph-to-graph translation model for property optimization. However, it requires a large number of translation pairs to train, which often involves expert human knowledge and is expensive to obtain. Others have used genetic/evolutionary algorithms to tackle this problem (Nigam et al., 2020; 2021a), which performs random mutations on chemical strings. Although these methods use learned discriminators to prune sub-optimal molecules, the random mutation process can become inefficient in searching for good molecules under complex property constraints. Xie et al. (2021); Fu et al. (2021) applies Markov Chain Monte Carlo (MCMC) sampling through editing molecules, while Kong et al. (2021) uses Bayesian optimization on the latent space. While there are extensive studies in exploration strategies for RL (Pathak et al., 2017; Burda et al., 2019; Ecoffet et al., 2021), diversity/novelty driven molecular generation is under-explored. We train a novelty/diversity-aware RL policy to search for novel, diverse molecules that retain desired properties. Our method initializes searches from model-discovered molecules, which greatly improves the efficiency and diversity of the generated molecules. Our ablation in Section 6 show that our proposed RL framework works well with the learned fragment vocabulary, while a simpler search strategy is not able to utilize this powerful editing paradigm.

## 3 PRELIMINARIES

**Message Passing Neural Networks (MPNN)** Molecules are represented as directed graphs, where the atoms are the nodes and the bonds are the edges of the graph. More formally, let $x = (V, E)$ denote a directed graph where $v_i \in V$ are the atoms, and $e_{ij} \in E$ are the edges of the graph. The network maintains hidden states $h^t_{e_{ij}}$ for each directed edge, where $t$ is the layer index. At each step, the hidden representations aggregate information from neighboring nodes and edges, and captures a larger neighborhood of atoms. Iteratively, the hidden states are updated as:

$$h_{e_{ij}}^{t+1} = f([h_{e_{ij}}^0; \sum_{k \in N(v_i) \setminus \{v_j\}} h_{e_{ki}}^t]) \tag{1}$$

Here, $f$ is parameterized by RNN cells (e.g. LSTM cells (Hochreiter & Schmidhuber, 1997) or GRU cells (Chung et al., 2014)), and $N(v_i)$ is the set of neighbors of $v_i$. After $T$ steps of message-passing, the final node embeddings $h_{v_i}$ are obtained by summing their respective incoming edge embeddings:

$$h_{v_i} = \text{ReLU}(W_o[h_{v_i}^0; \sum_{v_k \in N(v_i)} h_{e_{ki}}^T]) \tag{2}$$

The final node embeddings are then summed to get a graph embedding representation $h_x = \sum_{v_i} h_{v_i}$.

**Vector-Quantised Variational Autoencoders (VQ-VAE)** To learn useful representations of fragments, we employ the VQ-VAE architecture (van den Oord et al., 2017), which maps molecule fragment graphs to a discrete latent space through using categorical distributions for the prior and posterior. The VQ-VAE defines a dictionary of $k$ embedding elements, $[s_1, s_2, ...s_k] \in \mathbb{R}^{k \times l}$. Given an input $x$ (here the graph for a molecular fragment), let $z_e(x) \in \mathbb{R}^{d \times l}$ be the output of the encoder (a MPNN in our case). We define $l$ to be the same dimension for both encoder output embeddings $z_e(x)$ and dictionary embeddings $s_i$, because input $z_q(x)$ is computed by finding the $l_2$ nearest neighbor dictionary elements for each row of $z_e(x)$:

$$z_q(x)_i = s_k, \text{where } k = \arg\min_j ||z_e(x)_i - s_j||_2 \text{ for } i = 1, \dots, d \tag{3}$$

This embedding scheme allows us to represent each molecular fragment using a length $d$ vector, where each entry takes value from $\{1, \dots, k\}$ that corresponds to the dictionary embedding index for that row. The combinatorial vocabulary defined by the VQ-VAE has the capacity to represent $k^d$ distinct molecular fragments, which lifts the constraints of a limited generative span under a fixed fragment vocabulary.

Since the discretization step does not allow for gradient flow, gradients are passed through the network through approximating the gradient from the dictionary embeddings to the encoder embeddings. Additionally, there is a *commitment loss* that encourages the encoder to output embeddings that are similar to those in the dictionary (hence commitment). The total loss of the VAE is the following:

$$\mathcal{L} = \log p(x|z_q(x)) + \sum_i ||\text{sg}[z_e(x)_i] - s_{ij}||_2^2 + \beta \sum_i ||z_e(x)_i - \text{sg}[s_{ij}]||_2^2 \tag{4}$$

Where $s_{ij}$ is the closest dictionary element $s_j$ for the $z_e(x)_i$. Additionally, $\beta$ is a hyperparameter that controls for contribution of the commitment term, and sg represents the stop-gradient operator.

## 4 METHODS

**Molecular Optimization.** The goal of molecular optimization is to generate a set of high-quality molecules $\mathcal{C}$ (Constrained set) which satisfy or optimize a set of properties $P$. High novelty and diversity (detailed in Section 5) are also desired for de novo generation applications. We model the molecular optimization problem as a Markov decision process (MDP), defined by the 5-tuple $\{\mathcal{S}, \mathcal{A}, p, r, \rho_0\}$, where the state space $\mathcal{S}$ is the set of all possible molecular graphs. As an overview, our method introduces novel designs over the action space $\mathcal{A}$ and the transition model $p$ (Section 4.1) by utilizing a distributional fragment vocabulary, learned by a VQ-VAE. We define the reward and initial state distribution, $r$ and $\rho_0$ (Section 4.2) accordingly for specified tasks and to implement the proposed sequential translation generation scheme. An illustration of our model is in Figure 1.

### 4.1 LEARNING DISTRIBUTIONAL FRAGMENT VOCABULARY

**Molecular Fragments** are extracted from molecules in the ChEMBL database (Gaulton et al., 2012). For each molecule, we randomly sample fragments by extracting subgraphs that contain

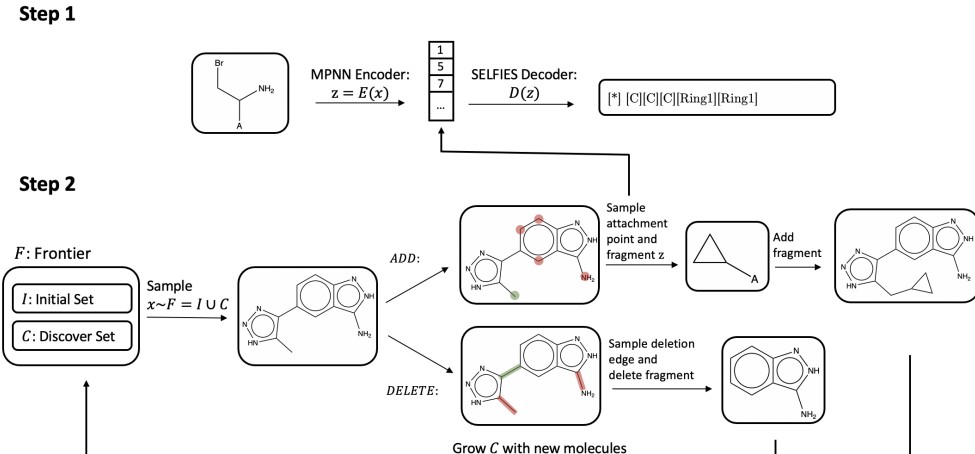

Figure 1: Overview of **Fra**gment-based **S**equential **T**ranslation (FaST). FaST is trained in a two-step fashion. In the first step, we train a VQ-VAE that embeds molecular fragments. In the second step, we train a search policy that uses the learned latent space as an action space. The search policy starts an episode by sampling a molecule from the *frontier* set $F$, which consists of an initial set of starting molecules ($\mathcal{I}$), and good molecules discovered by the policy ($\mathcal{C}$). The molecule is encoded by an MPNN, which is then used to predict either an *Add* or *Delete* action. When the *Add* action is selected, the model predicts and samples an atom as the attachment point and subsequently predicts a fragment to attach to that atom. When the *Delete* action is selected, the model samples a directed edge, indicating the molecular fragment to be deleted.

ten or fewer atoms that have a single bond attachment to the rest of the molecule. We then use a VQ-VAE to encode these fragments into a meaningful latent space. The use of molecular fragments simplifies the search problem, while the variable-sized fragment distribution maintains the reachability of most molecular compounds. Because our search algorithm ultimately uses the latent representation of the molecules as the action space, we find that using a VQ-VAE with a categorical prior instead of the typical Gaussian prior makes RL training stable and provides good performance gains (Tang & Agrawal, 2020; Grill et al., 2020). The training instability under a normal VAE with Gaussian prior and continuous latents causes failture of the RL training. Our ablation study also shows that the fragment samples from a VQ-VAE are more diverse than the samples from a continuous VAE (Section 6).

**Encoder/Decoder** We use MPNN encoders for any graph inputs, which include both fragments for the VQ-VAE, as well as molecular states during policy learning. The graph models are especially suitable for describing actions on the molecular state, as they explicitly parametrize the representations of each atom and bond. Meanwhile, the decoder architecture is a recurrent network that decodes a SELFIES representation of a molecule. We choose a recurrent network for the decoder because we do not need the full complexity of a graph decoder. Due to the construction scheme, the fragments are rooted trees, and all have a single attachment point. As our fragments are small in molecular size ($\leq$ 10 atoms), the string grammar is simple to learn, and we find the SELFIES decoder works well empirically (see Appendix F for more details).

**Adding and deleting fragments as actions.** At each step of the MDP, the policy network first takes the current molecular graph as input and produces a Bernoulli distribution on whether to add or delete a fragment. Equipped with the fragment VQ-VAE, we define the *Add* and *Delete* actions at the fragment-level:

- **Fragment Addition.** The addition action is characterized by (1) a probability distribution over the atoms of the molecule: $p_{add}(v_i) = \sigma[\text{MLP}(h_v)]$, where $\sigma$ is the softmax operator. (2) Conditioned on the graph embedding $h_x$ and the attachment point atom $v_{add}$ sampled from $p_{add}$, we predict a $d$-channel categorical distribution $p_{fragment} = \sigma[\text{MLP}([h_{v_{add}}; h_x])] \in \mathbb{R}^{d \times k}$, where each row of $p_{fragment}$ sums to 1. We can then sample the discrete categorical latent $z_{add} \in \{1, ..., k\}^d$ from $p_{fragment}$. The fragment to add

is then obtained by deocoding $z_{add}$ through the learned frozen fragment decoder. We then assemble the decoded fragment with the current molecular graph by attaching the fragment to the predicted attachment point $v_{add}$. Note that the attachment point over the fragment is indicated through the generated SELFIES string.

- **Fragment Deletion.** The deletion action acts over the directed edges of the molecule. A probability distribution over deletable edges is computed with a MLP: $p_{del}(e_{ij}) = \sigma[\text{MLP}(h_{e_{ij}})]$. One edge is then sampled and deleted; since the edges are directed, the directionality specifies the the molecule to keep and the fragment to be deleted.

With the action space $\mathcal{A}$ defined as above, the transition model for the MDP is simply $p(s'|s, a) = 1$ if applying the addition/deletion action $a$ to the molecule $s$ results in the molecule $s'$, and $p(s'|s, a) = 0$ otherwise. The fragment-based action space is powerful and suitable for policy learning as it (1) is powered by the enormous distributional vocabulary learned by the fragment VQ-VAE, thus spans a diverse set of editing operations over molecular graphs; (2) exploits the meaningful latent representation of fragments since the representation of similar fragments are grouped together. These advantages greatly simplify the molecular search problem. We terminate an episode when the molecule fails to satisfy the desired property or when the episode exceeds ten steps.

## 4.2 DISCOVER NOVEL MOLECULES THROUGH SEQUENTIAL TRANSLATION

We propose sequential translation that incrementally grows the set of discovered novel molecules and use the model-discovered molecules as starting points for further search episodes. This regime of starting exploration from states reached in previous episodes was also explored under the setting of RL from image inputs (Ecoffet et al., 2021). More concretely, we implement sequential translation with a reinforcement learning policy that operates under the fragment-based action space defined in Section 4.1, while using a moving initial state distribution $\rho_0$, which is a distribution over molecules in the *frontier* set $\mathcal{F} = \mathcal{I} \cup \mathcal{C}$. By starting new search episodes from the frontier set – the union of the initial set and good molecules that are discovered by the RL policy, we achieve efficient search in the chemical space by staying close to the high-quality subspace and achieve novel molecule generation through a sequence of fragment-based editing operations to the known molecules. Our proposed search algorithm is detailed in Algorithm 1.

**Discover novel molecules and expand the frontier.** Our method explores the chemical space with a property-aware and novelty/diversity-aware reinforcement learning policy that proposes addition/deletion modifications to the molecular state at every environment step to optimize for the reward $r$. We gradually expand the discovered set $\mathcal{C}$ by adding *qualified* molecules found in the RL exploration within the MDP. A molecule $x$ is qualified if: (1) $x$ satisfies the desired properties measured by property scores

$$C_P(x) = \prod_{p \in P} \mathbb{1}\{\texttt{score}_p(x) > \texttt{threshold}_p\} \tag{5}$$

where $P$ is the set of desired properties and $\texttt{threshold}_p$ is the score threshold for satisfying property $p$. A molecule $x$ satisfying all desired properties has $C_P(x) = 1$ and $C_P(x) = 0$ otherwise. (2) $x$ is novel/diverse compared to molecules currently in the frontier $F$, measured by fingerprint similarity (detailed in Section 5):

$$C_{ND}(x) = \mathbb{1}\{\max_{i \in I} \texttt{sim}(x, i) < \texttt{threshold}_{nov}\} \cdot \mathbb{1}\{\operatorname*{mean}_{g \in G}(\texttt{sim}(x, g)) < \texttt{threshold}_{div}\} \tag{6}$$

Where $\texttt{sim}$ denotes fingerprint similarity, $\texttt{threshold}_{nov}$ and $\texttt{threshold}_{div}$ are predefined similarity thresholds for novelty and diversity, $\mathcal{I}$ and $\mathcal{C}$ are the initial set of good molecules and model discovered molecules as defined in previous sections. A molecule that satisfies novelty/diversity criterion has $C_{ND}(x) = 1$ and $C_{ND}(x) = 0$ otherwise.

We use a reward of $+1$ for a transition that results in a molecule qualified for the set $\mathcal{C}$, and discourage the model from producing invalid molecules by adding a reward of $-0.1$ for a transition that produces an invalid molecular graph [1]:

$$r(x, a) = C_P([x \leftarrow a]) \cdot C_{ND}([x \leftarrow a]) - 0.1 \cdot \mathbb{1}([x \leftarrow a] \text{ invalid}) \tag{7}$$

where $[x \leftarrow a]$ denotes the molecule resulting from editing $x$ with the fragment addition/deletion action $a$.

---

[1] validity is checked by the chemistry software RDKit.

---

**Algorithm 1** Molecular Optimization through **Fra**gment-based **S**equential **T**ranslation (FaST)

---

1: Input $N$ the desired number of discovered new molecules
2: Input $\mathcal{I}$ the initial set of molecules
3: Input $D$ the pretrained fragment decoder of VQ-VAE
4: Input $C_P : \mathcal{S} \to \{0, 1\}$ returns 1 if the input $x$ satisfies desired properties $\quad\quad\quad$ ▷ Equation (5)
5: Input $C_{ND} : \mathcal{S} \to \{0, 1\}$ returns 1 if the input $x$ satisfies novelty/diversity criterion $\quad$ ▷ Equation (6)
6: Let $\mathcal{C} = \emptyset$ be the discovered set of molecules
7: Let $\mathcal{F} = \mathcal{I} \cup \mathcal{C}$ be the frontier where search is initialized from
8: Let $t = 0$ be the number of episodes
9: **while** $|\mathcal{C}| \leq N$ **do**
10: $\quad$ Let $t = t + 1$
11: $\quad$ Update $UCB(x_0, t) \forall x_0 \in \mathcal{F}$ according to Equation (8)
12: $\quad$ Sample initial molecule $x = (V, E)$ from $p_{init} = \sigma[UCB(x_0, t)] \forall x_0 \in \mathcal{F}$
13: $\quad$ Let $\texttt{step} = 0$
14: $\quad$ **while** $C_P(x) = 1$ `&` $\texttt{step} < \texttt{T}$ **do** $\quad\quad\quad\quad\quad\quad\quad\quad$ ▷ T $= 10$ in our experiments
15: $\quad\quad$ Encode $x$ with MPNN$(x)$ to get node representation $h_v, \forall v \in V$ and graph representation $h_x$
16: $\quad\quad$ Sample action type $a \in \{\text{ADD}, \text{DELETE}\}$ from $p_{action} = \sigma[\text{MLP}(h_x)]$
17: $\quad\quad$ **if** $a = \text{ADD}$ **then**
18: $\quad\quad\quad$ Sample $v_{add}$ from $p_{add}(v) = \sigma[\text{MLP}(h_v)] \; \forall v \in V$
19: $\quad\quad\quad$ Sample fragment encoding $z_{add}$ from $p_{fragment} = \text{MLP}([h_x; h_{v_{add}}])$ $\quad\quad$ ▷ Section 4.1
20: $\quad\quad\quad$ Decode fragment $y = D(z_{add})$
21: $\quad\quad\quad$ Add fragment $y$ to molecule: $x \leftarrow x + y$
22: $\quad\quad$ **else**
23: $\quad\quad\quad$ Sample $e$ from $p_{del}(e) = \sigma[\text{MLP}(h_{e_{ij}})] \; \forall e \in E$ $\quad\quad\quad\quad\quad\quad$ ▷ Section 4.1
24: $\quad\quad\quad$ Let $y$ be the fragment designated by $e$, delete fragment $x \leftarrow x - y$
25: $\quad\quad$ **if** $C_P(x) = 1$ `&` $C_{ND}(x) = 1$ **then**
26: $\quad\quad\quad$ $\mathcal{C} \leftarrow \mathcal{C} \cup \{x\}$
27: $\quad\quad\quad$ $\mathcal{F} \leftarrow \mathcal{I} \cup \mathcal{C}$
28: $\quad\quad$ Let $\texttt{step} \leftarrow \texttt{step} + 1$

---

**Initialize search episodes from promising candidates.** To bias the initial state distribution $\rho_0$ to favor molecules that can derive more novel high-quality molecules, we keep an upper-confidence-bound (UCB) score for each initial molecule in the frontier $F$. We record the number of times we initiate a search $N(x, t)$ from a molecule $x \in F$, and the number of molecules qualified for adding to $\mathcal{C}$ that is found in an episode strating from $x$: $R(x, t)$. Here $t = \sum_{x \in \rho_0} N(x)$ is the total number of search episodes. The UCB score of the initial molecule $m$ is calculated by:

$$UCB(x, t) = \frac{R(x, t)}{N(x, t)} + \frac{\sqrt{\frac{3}{2} \log(t + 1)}}{N(x, t)} \quad\quad\quad\quad (8)$$

The probability of a molecule in the initialization set being sampled as the starting point of a new episode is then computed by a softmax over the UCB scores: $p_{init}(x, t + 1) = \frac{\exp(UCB(x,t))}{\sum_{x \in I} \exp(UCB(x,t))}$.

To summarize, FaST learns a policy that (1) choose good initial molecules to start search episodes; (2) choose to add a fragment to or delete a subgraph from a given state (a molecule in our case); (3) choose what to add through predicting a fragment latent embedding, or what to delete through predicting a directed edge, and remove part of the molecular graph accordingly.

Although we present our method in this section under the most realistic multi-objective optimization task settings (with experiments in Section 5), our method is easily extendable to other problem settings by modifying the constraints $C_P$, $C_{ND}$, and the reward function $r$ accordingly. For example, see Appendix B for the application of our method to the standard constrained penalized $\log P$ task and Section 5 for multi-objective molecular optimization under different novelty/diversity metrics.

## 5 EXPERIMENTS

**Datasets.** We use benchmark datasets for molecular optimization, which aims to generate ligand molecules for inhibition of two proteins: glycogen synthase kinase-3 beta (GSK3$\beta$) and c-Jun N-terminal kinase 3 (JNK3). Following previous work (Jin et al., 2020; Xie et al., 2021; Nigam et al., 2021a), we adopt the same strategy of using a random forest trained on these datasets as the oracle property predictor, and incorporate the additional factors, quantitative estimate of drug-likeness (QED) (Bickerton et al., 2012) and synthetic accessibility (SA) (Ertl & Schuffenhauer, 2009) as our

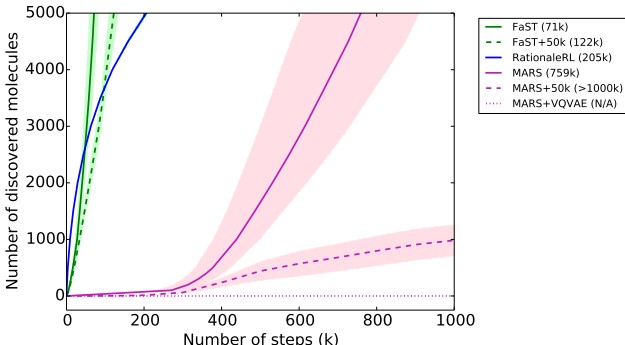

Figure 2: Comparison of the sample complexity of different methods. The x-axis is the number of molecules searched through and the y-axis is the number of discovered molecules, where the target is to obtain a set of 5,000 molecules that achieves **SR** = 1, **Nov** = 1 and **Div** = .7. Our method (FaST) achieves the best sample complexity with 71k molecules visited. Fast+50k and MARS+50k are their respective models trained with the same fixed fragment vocabulary extracted from ChEMBL.

optimization objectives. Single property optimization is often a flawed task, because the generator can overfit to the pretrained predictor and generate unsynthesizable compounds.

**Evaluation metrics.** Following previous works, we evaluate our generative model on three target metrics, success, novelty and diversity. 5,000 molecules are generated by the model, and the metric scores are computed as follows: **Success rate (SR)** measures the proportion of generated molecules that fit the desired properties. **Novelty (Nov)** measures how different the generated molecules are compared to the set of actives in the dataset (range $[0, 1]$), and **Diversity (Div)** measures how different the generated molecules are compared to each other (range $[0, 1]$). **PM** is the product of the three metrics above (**PM** = **SR** · **Nov** · **Div**).

**Implementation details.** We construct the initial set of molecules for our search algorithm from the rationales extracted from Jin et al. (2020). These rationales are obtained through a sampling process on the active molecules that tries to minimize the size of the rationale subgraph, while maintaining their inhibitory properties. Rationales for multi-property tasks (GSK3$\beta$+JNK3) are extracted by combining the rationales for single-property tasks. Initializing generation with subgraphs is commonly done in molecular generative models such as Shi et al. (2020) and Kong et al. (2021). We train the RL policy using the Proximal Policy Optimization (PPO, Schulman et al. 2017) algorithm. We find the RL training robust despite both the reward function $r$ and the initial state distribution $\rho_0$ are non-stationary (i.e., changing during RL training). Hyperparameters used for producing the results in Section 5 and molecule samples from FaST are included in Appendix D and Appendix E.

**Baseline methods. Rationale-RL** (Jin et al., 2020) extracts rationales of the active molecules and then uses RL to train a completion model that add atoms to the rationale in a sequential manner to generate molecules satisfying the desired properties. **GA+D & JANUS** (Nigam et al., 2020; 2021a) are two genetic algorithms that use random mutations of SELFIES strings to generate promising molecular candidates; JANUS leverages a two-pronged approach, accounting for mutations towards both exploration and exploitation. **MARS** (Xie et al., 2021) uses Markov Chain Monte Carlo (MCMC) sampling to iterative build new molecules by adding or removing fragments, and the model is trained to fit the distribution of the active molecules. To provide a fair comparison against baselines that do not use rationales, we additionally include a baseline **MARS+Rationale** that initialize the MARS algorithm with the same starting initial rationale set used in Rationale-RL and our method. where possible, we use the numbers from the original corresponding paper.

**Performance.** The evaluation metrics are shown in Table 1; FaST significantly outperforms all baselines on all tasks including both single-property and multi-property optimization. On the most challenging task, GSK3$\beta$+JNK3+QED+SA, FaST improves upon the previous best model by over 30% in the product of the three evaluation metrics. Our model is able to efficiently search for molecules that stay within the constrained property space, and discover novel and diverse molecules by sequentially translating known and discovered active molecules. The MARS+Rationale model, which uses the same rationales as the initialization for their search algorithm, does not perform

Table 1: FaST outperforms all baselines on both single-property and multi-property optimization. Error bars indicates one standard deviation, obtained from averaging 5 random seeds.

| Model | GSK3$\beta$ | | | | GSK3$\beta$+QED+SA | | | |
|---|---|---|---|---|---|---|---|---|
| | SR | Nov | Div | PM | SR | Nov | Div | PM |
| Rationale-RL | 1.00 | .534 | .888 | .474 | .699 | .402 | .893 | .251 |
| GA+D | .846 | 1.00 | .714 | .600 | .891 | 1.00 | .628 | .608 |
| JANUS | 1.00 | .829 | .884 | .732 | - | - | - | - |
| MARS | 1.00 | .840 | .718 | .600 ($\pm$ .04) | .995 | .950 | .719 | .680 ($\pm$ .03) |
| MARS+Rationale | .995 | .804 | .746 | .597 ($\pm$ .07) | .981 | .800 | .807 | .632 ($\pm$ .07) |
| FaST | 1.00 | 1.00 | .905 | **.905 ($\pm$ .000)** | 1.00 | 1.00 | .861 | **.861 ($\pm$ .001)** |

| Model | JNK3 | | | | JNK3+QED+SA | | | |
|---|---|---|---|---|---|---|---|---|
| | SR | Nov | Div | PM | SR | Nov | Div | PM |
| Rationale-RL | 1.00 | .462 | .862 | .400 | .623 | .376 | .865 | .203 |
| GA+D | .528 | .983 | .726 | .380 | .857 | .998 | .504 | .431 |
| JANUS | 1.00 | .426 | .895 | .381 | - | - | - | - |
| MARS | .988 | .889 | .748 | .660 ($\pm$ .04) | .913 | .948 | .779 | .674 ($\pm$ .02) |
| MARS+Rationale | .976 | .843 | .780 | .642 ($\pm$ .04) | .634 | .779 | .787 | .386 ($\pm$ .08) |
| FaST | 1.00 | 1.00 | .905 | **.905 ($\pm$ .001)** | 1.00 | .866 | .856 | **.741 ($\pm$ .001)** |

| Model | GSK3$\beta$+JNK3 | | | | GSK3$\beta$+JNK3+QED+SA | | | |
|---|---|---|---|---|---|---|---|---|
| | SR | Nov | Div | PM | SR | Nov | Div | PM |
| Rationale-RL | 1.00 | .973 | .824 | .800 | .750 | .555 | .706 | .294 |
| GA+D | .847 | 1.00 | .424 | .360 | .857 | 1.00 | .363 | .311 |
| JANUS | 1.00 | .778 | .875 | .681 | 1.00 | .326 | .821 | .268 |
| MARS | .995 | .753 | .691 | .520 ($\pm$ .08) | .923 | .824 | .719 | .547 ($\pm$ .05) |
| MARS+Rationale | .976 | .843 | .780 | .642 ($\pm$ .04) | .654 | .687 | .724 | .321 ($\pm$ .09) |
| FaST | 1.00 | 1.00 | .863 | **.863 ($\pm$ .001)** | 1.00 | 1.00 | .716 | **.716 ($\pm$ .011)** |

well compared to the original implementation, which initializes each search with a simple "C-C" molecule.

**Sample complexity comparison given performance thresholds.** Another comparison scheme is to let a model keep generating molecules until it achieves a good candidate set under certain performance thresholds. Under this evaluation protocol, all models will have the same or very similar performance in SR, Nov, and Div. The metric of interest will then be the sample complexity of the algorithm: how many molecules it requires to visit/generate to obtain a good candidate set. This places every model under the same regime, allowing each model to generate molecules in a novelty/diversity-aware setting. We compare FaST to Rationale-RL and MARS under this setting in Figure 2, where we impose SR=1, Nov=1, Div=.7 for the candidate set. FaST on average searched through 71k molecules in total to gather the 5k proposal set, while Rational-RL and MARS need to search through 205k and 759k molecules to obtain their corresponding proposal sets. Being a pretrained generative model, Rationale-RL has a steeper slope initially but then slows down to find more good molecules. The flexibility of our learned vocabulary and the RL search strategy lead to the superior performance of FaST, which we further verify through ablation study in Section 6.

**Optimize for different novelty/diversity metrics.** The Morgan fingerprints used for similarity comparison contain certain inductive biases. Under different applications, different novelty/diversity metrics may be of interest. To demonstrate the viability of our model under any metrics, we train FaST using Atom Pairs (AP) fingerprints (Carhart et al., 1985) on the GSK3$\beta$+JNK3+QED+SA task. The results, and discussion of the different fingerprint methods, are reported in Appendix C. We find that (1) FaST can still find high-quality molecules that are novel and diverse, while the baseline methods suffer a low novelty; (2) FaST trained with AP fingerprints still attains good performance when evaluated under Morgan fingerprints but the reverse is not true. This result shows that Novelty/diversity under AP fingerprints is a stricter criterion to satisfy and necessitates novelty/diversity-awareness during optimization.

**Penalized logP Maximization.** To demonstrate the wide applicability of our method to any molecular optimization task, we also include our results on the standard constrained penalized logP optimization task in Appendix B. We show that our model significantly outperforms all baselines in this task under different constraint levels. We also provide insight on the task itself: while this task has

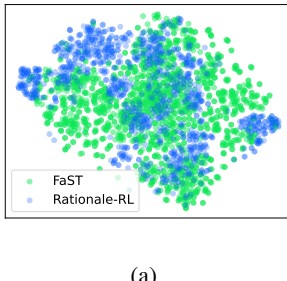 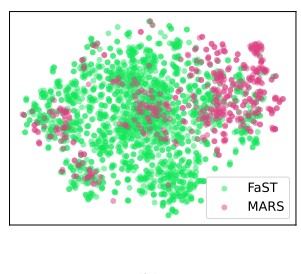

(a)                                                    (b)

Figure 3: (a, b) plots the t-SNE embedding of fragments from generated compounds of our model vs. Rational-RL and MARS. The visualization shows that that our model produces a much more diverse set of fragments, which is a proxy for functional groups appearing in generated molecules.

been studied in many previous works, the task, as it is currently defined, is not entirely chemically meaningful. Additionally, one drawback of this task is that a model can achieve high performance by simply generating large molecules. A detailed discussion can be found in Appendix B.

## 6    ABLATION AND ANALYSIS

**Diversity of generation.** In addition to the fingerprint diversity metrics presented in Section 5, we also examine functional group diversity. We extract all unique molecular fragments of the 5,000 molecules generated for GSK3$\beta$+JNK3+QED+SA task for each model, and produce t-SNE visualization of these fragments in Figure 3a and Figure 3b. In total, we extracted 1.7k unique fragments from our model outputs vs only 1.1k unique fragments for Rational-RL and 500 unique fragments from MARS. The visualization shows that the fragments in the molecules generated by our model spans a much larger chemical space. This confirms the advantages of using a learned vocabulary, compared to using a fixed set of fragments, as we are able to utilize a much more diverse set of chemical subgraphs. Sampled trajectories (Figure 6) and molecules (Figure 7) are included in Appendix E.

**Benefit of distributional vocabulary.** To investigate the benefit of using a distributional vocabulary, instead of using the pretrained VQ-VAE, we also train our model using a fixed vocabulary of fragments, which consists of roughly 50k unique fragments (the same set used to pretrain the VQ-VAE). Figure 2 compares the performance of the two models. On average, the model with fixed fragments took 122k steps, while with VQ-VAE it only took 71k steps to find a set of 5,000 good molecules (72% improvement). We further analyze the benefit of using discrete latents with a VQ-VAE rather than continuous latents with a Gaussian prior VAE in Appendix A.

**Importance of RL search.** We also demonstrate the importance of our RL search policy compared to previous sampling methods. To do so, we run MARS with both the 50k fixed fragments and our VQ-VAE. Figure 2 shows the performance of using the 50k fixed fragment vocabulary compared to the original MARS model which uses a small 1k vocabulary. When the vocabulary is large, MARS exhibits very poor sample complexity. Additionally, we also implemented our VQ-VAE with the sampling strategy proposed in MARS, but this model was altogether unable to successfully generate good candidate molecules. Therefore, we see that when the vocabulary is more complex, we need a better search strategy, highlighting the importance of our RL algorithm.

## 7    CONCLUSION

We propose a new framework for molecular optimization, which leverages a learned vocabulary of molecular fragments to search the chemical space efficiently. We demonstrate that **Fra**gment-based **S**equential **T**ranslation (FaST), which adaptively grows a set of promising molecular candidates, can generate high-quality, novel, and diverse molecules on single-property and multi-property optimization tasks. Ablation study shows that all components of our proposed method contribute to its superior performance. The learning of a flexible vocabulary is a complementary module to other research in fragment-based drug design. Incorporating FaST to more practical drug discovery pipelines while taking synthesis paths in mind is an exciting avenue for future work.

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

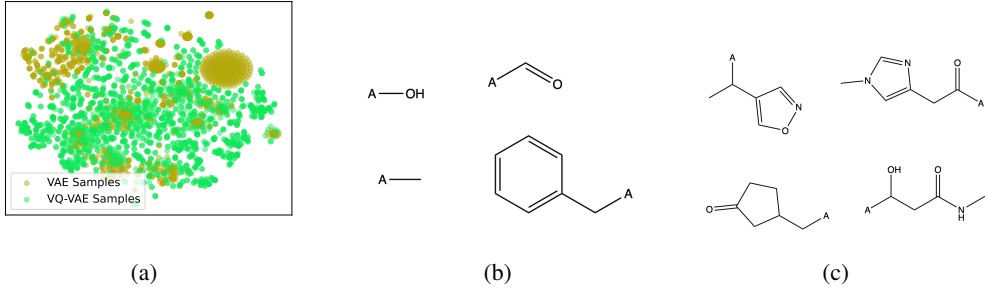

(a)           (b)           (c)

Figure 4: (a) t-SNE of fragments sampled from a trained VAE and VQ-VAE. The fragments sampled from the VAE are tightly clustered, showing much less diversity compared to the fragments sampled from the VQ-VAE. (b,c) Random samples from the VAE (b) and the VQ-VAE (c). "A" denotes the attachment point for the fragment. We see that the samples from the VAE are relatively simple. Meanwhile, the samples from the VQ-VAE are more diverse.

## A  VOCABULARY LEARNING THROUGH VQ-VAE

To evaluate the benefits of VQ-VAE over a typical VAE trained with Gaussian priors, we train both models, and look at the distribution of fragments. Figure 4a compares the t-SNE distributions of the two models, where we sample 2,000 fragments from each model. The VAE model has tight clusters, while the VQ-VAE model exhibits a much more diverse set of fragments. We visualize random samples from VAE and VQ-VAE Figure 4, where we see that the samples from VAE are relatively simple and generic fragments, while samples from the VQ-VAE demonstrate diverse patterns. This is because the more generic fragments appear more frequently in real molecules, and a Gaussian prior over the fragment latent space would favor these fragments.

## B  CONSTRAINED PENALIZED LOGP TASK

To demonstrate the general applicability of our model for any molecular optimization task, we also run our model on another constrained optimization task, here optimizing for penalized octanol-water partition coefficients (logP) scores of ZINC (Irwin et al., 2012) molecules. The penalized logP score is the logP score penalized by synthetic accessibility and ring size. We use the exact computation in You et al. (2018), where the components of the penalized logP score are normalized across the entire 250k ZINC training set. The generated molecules are constrained to have similar Morgan fingerprints (Rogers & Hahn, 2010) as the original molecules.

Following the same setup as previous work (Jin et al., 2019b; You et al., 2018; Shi et al., 2020; Nigam et al., 2020; Kong et al., 2021), we try to optimize the 800 test molecules from ZINC with the lowest penalized logP scores (the initial set $\mathcal{I}$). Specifically, the task is to translate these molecules into new molecules with the Tanimoto similarity of the fingerprints constrained within $\delta \in \{.4, .6\}$. This task aims for optimizing a certain quantity (instead of satisfying property constraints) and is a translation task (need to stay close to original molecules rather than finding novel ones). To run FaST on this task, we apply the following changes to the reward function, the qualification criterion, and the episode termination criterion, of FaST. We denote $\texttt{score}(x)$ to be the penalized logP scoring function, and $\texttt{sim}(\cdot, \cdot)$ to be the Tanimoto similarity between two molecules:

- reward $r = \texttt{score}(x_j) - \texttt{score}(x_i)$ for any transition from molecule $x_i \rightarrow x_j$
- $\mathcal{C}$, the discovered set contains all explored molecules that satisfy Equation (5), where the threshold is given by the input parameter $\delta$
- We terminate an episode when the number of steps exceeds 10.

For each molecule we add to $G$, we keep track of its original parent (the molecule from the 800 test molecules). After training, for each of the 800 test molecules, we take the set of translated molecules in $G$, and select the one with the highest property score.

Table 2: Results on the constrained penalized logP task. FaST significantly outperform all baselines.

| Method | $\delta = 0.4$ | | $\delta = 0.6$ | |
|---|---|---|---|---|
| | Improvement | Success | Improvement | Success |
| JT-VAE (Jin et al., 2018) | $0.84 \pm 1.45$ | 83.6 % | $0.21 \pm 0.71$ | 46.4 % |
| GCPN (You et al., 2018) | $2.49 \pm 1.30$ | 100.0 % | $0.79 \pm 0.63$ | 100.0 % |
| DEFactor (Assouel et al., 2018) | $3.41 \pm 1.67$ | 85.9 % | $1.55 \pm 1.19$ | 72.6 % |
| MolDQN (Zhou et al., 2019) | $3.37 \pm 1.62$ | 100 % | $1.86 \pm 1.21$ | 100 % |
| GraphAF (Shi et al., 2020) | $3.74 \pm 1.25$ | 100 % | $1.95 \pm 0.99$ | 100 % |
| GP-VAE (Kong et al., 2021) | $4.19 \pm 1.30$ | 98.9 % | $2.25 \pm 1.12$ | 90.3 % |
| VJTNN (Jin et al., 2019b) | $3.55 \pm 1.67$ | - | $2.33 \pm 1.17$ | - |
| GA+D (Nigam et al., 2020) | $5.93 \pm 1.14$ | 100 % | $3.44 \pm 1.09$ | 99.8 % |
| FaST (Ours) | $18.09 \pm 8.72$ | 100 % | $8.98 \pm 6.31$ | 96.9 % |

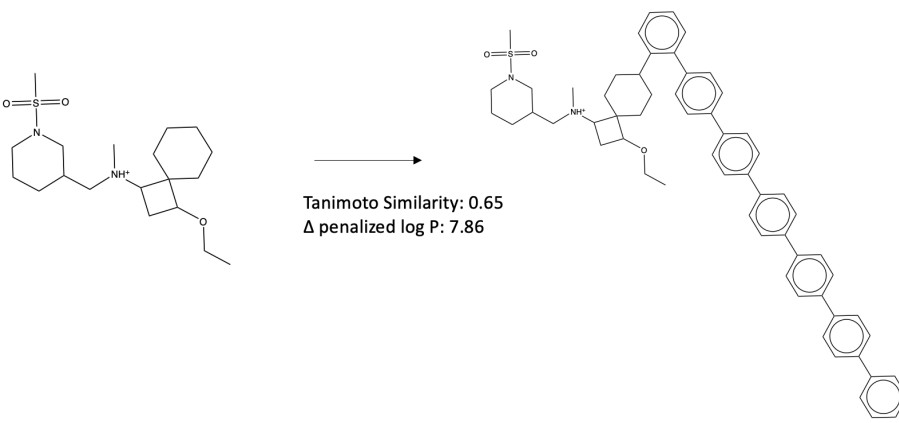

Tanimoto Similarity: 0.65
Δ penalized log P: 7.86

Figure 5: Sample translation of our model for the constrained penalized logP task ($\delta = 0.6$). The model generates a molecule with repeating aromatic rings; though not realistic, this molecule achieves a high score, while having close Tanimoto similarity using Morgan fingerprints.

Results are shown in Table 2; our method greatly outperforms the other baselines, but we point out a few flaws intrinsic to the task. Because the similarity is computed through Morgan fingerprint, which are hashes of substructures, repeatedly adding aromatic rings can often not change the fingerprint by a lot. Nevertheless, adding aromatic rings will linearly increase penalized logP score, which allows trivial solutions to produce high scores for this task (see Figure 5). This phenomenon is noted by Nigam et al. (2020), but they add a regularizer to constrain the generated compounds to look similar to the reference molecules. Due to the mentioned issues, we believe this task can be reformulated. For instance, one could use a different fingerprint method so that the fingerprint similarity is not so easily exploited (see AP (Carhart et al., 1985), MACCS (Durant et al., 2002), or ROCS (Hawkins et al., 2010)), or size constraints should be incorporated. Nevertheless, we provide our results for comparison to other molecular generation methods.

In general, the task of optimizing (increasing) the penalized logP scores is not entirely meaningful. According to Lipinski's rule of five (Lipinski et al., 1997), which are widely established rules to evaluate the druglikeness of molecules, the logP score should be lower than 5. So an unbounded optimization of logP has little practical usability. Perhaps a better task would be to optimize for all 5 rules in Linpinski's rule of five which includes constraints involving the number of hydrogen bond donors/acceptors and molecular mass.

## C  DIFFERENT NOVELTY/DIVERSITY METRICS

FaST is capable of optimizing for different novelty/diversity metrics. In this section, we compute the novelty/diversity metrics using atom-pair (AP) fingerprints (Carhart et al., 1985). While Morgan fingerprints have successfully been applied to many molecular tasks such as drug screening, it has

some failure modes (Capecchi et al., 2020). Namely, Morgan fingerprints is often not informative about the size or the shape of the molecules. These properties are better captured in AP fingerprints, as AP fingerprints account for all atom pairs, including their pairwise distances. We run the same experiment on the GSK3$\beta$+JNK3+QED+SA task described in Section 5, but change the fingerprint from Morgan to AP for the novelty/diversity metrics. The results are shown in Table 3 with comparison to baselines. We observe that our method outperform baselines by a greater margin, especially in the novelty metric. This is not surprising because our model can explicitly optimize for any similarity metric, while the baseline methods are not novelty/diversity-aware during training. Interestingly, we find that optimizing for AP fingerprints also yields molecules that score high under Morgan fingerprints for this task (but the converse is not true).

Table 3: Results on the GSK3$\beta$+JNK3+QED+SA task using AP fingerprints instead of Morgan fingerprints for novelty/diversity computation.

| Method | Success (SR) | | Novelty (Nov) | | Diversity (Div) | |
|---|---|---|---|---|---|---|
| | Morgan | AP | Morgan | AP | Morgan | AP |
| Rationale-RL | .750 | .750 | .555 | .023 | .706 | .630 |
| MARS | .923 | .733 | .824 | .077 | .719 | .644 |
| FaST (Morgan) | 1.00 | 1.00 | 1.00 | .555 | .716 | .674 |
| FaST (AP) | 1.00 | 1.00 | .987 | .867 | .675 | .719 |

## D   IMPLEMENTATION DETAILS

Table 4: Hyperparameters for the VQ-VAE.

| VQ-VAE Param | Value |
|---|---|
| hidden size | 200 |
| MPNN depth | 4 |
| MPNN output size ($d$) | 10 |
| # Dictionary elements ($k$) | 10 |
| Dictionary latent size ($l$) | 10 |
| batch size | 32 |
| dictionary loss coef | 1.0 |
| commitment loss coef | 1.0 |
| learning rate | 1e-4 |
| # epochs | 10 |

Table 5: Hyperparameters for training the RL Agent.

| | Param Name | Value |
|---|---|---|
| | learning rate | 2e-4 |
| | $\gamma$ | 0.999 |
| | $\lambda_{GAE}$ | 0.95 |
| | batch size | 64 |
| Agent | PPO Epoch | 3 |
| | param clip | 0.2 |
| | value loss coef | 0.5 |
| | entropy loss coef | 0.01 |
| | $\epsilon$ | 1e-5 |
| | max grad norm | 0.5 |
| Actor | hidden size | 1024 |
| | hidden depth | 1 |
| Critic | hidden size | 256 |
| | hidden depth | 1 |

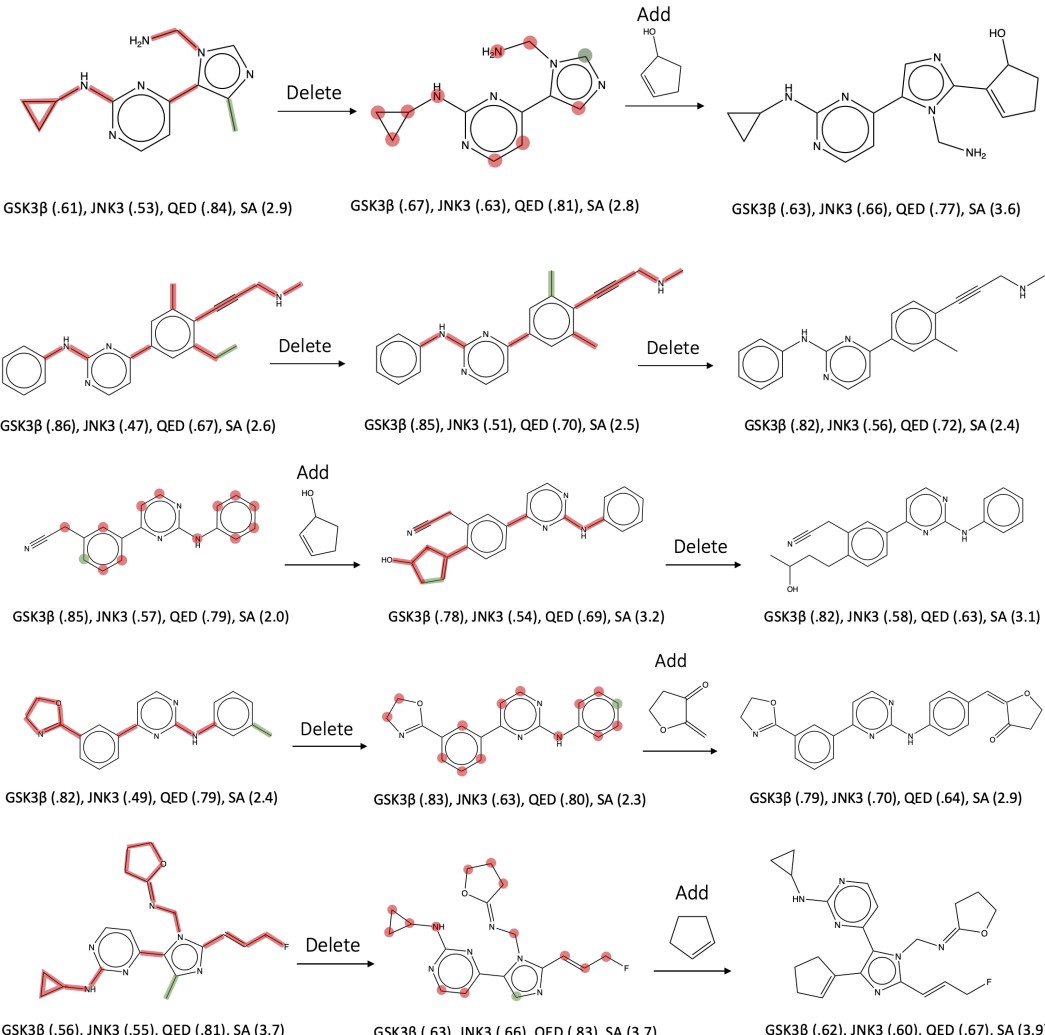

Figure 6: Samples of FaST on the GSK3$\beta$+JNK3+QED+SA task. Red indicates the atoms and bonds that are eligible for addition/deletion, while green indicates the selected atom or bond.

## E    SAMPLE TRAJECTORIES AND OPTIMIZED MOLECULES FROM FAST

We provide more example molecular optimization trajectories of our model on the GSK3$\beta$+JNK3+QED+SA task in Figure 6.

Additional generated moles for the GSK3$\beta$+JNK3+QED+SA task are shown in Figure 7

## F    DECODER ARCHITECTURE: SELFIES VS SMILES

For our VAE, we decode into string representations of fragments for several reasons. **(1)** Graph decoders are complex and requires a lot of hand engineered rules to work. In fact, many graph decoders require their own vocabularies, which makes the decoder architecture even more complex. **(2)** Our fragments are small, less than 10 atoms, so a simpler representation of the fragments in string form is appropriate. Additionally, we choose to use SELFIES strings over SMILES strings, because SELFIES strings have a more robust grammar compared to SMILES strings, and has been shown through many past experiments (Krenn et al., 2020). Empirically, we see in Figure 8 that SELFIES strings gives better reconstruction accuracy compared to using SMILES strings for the decoder in the VAE architecture.

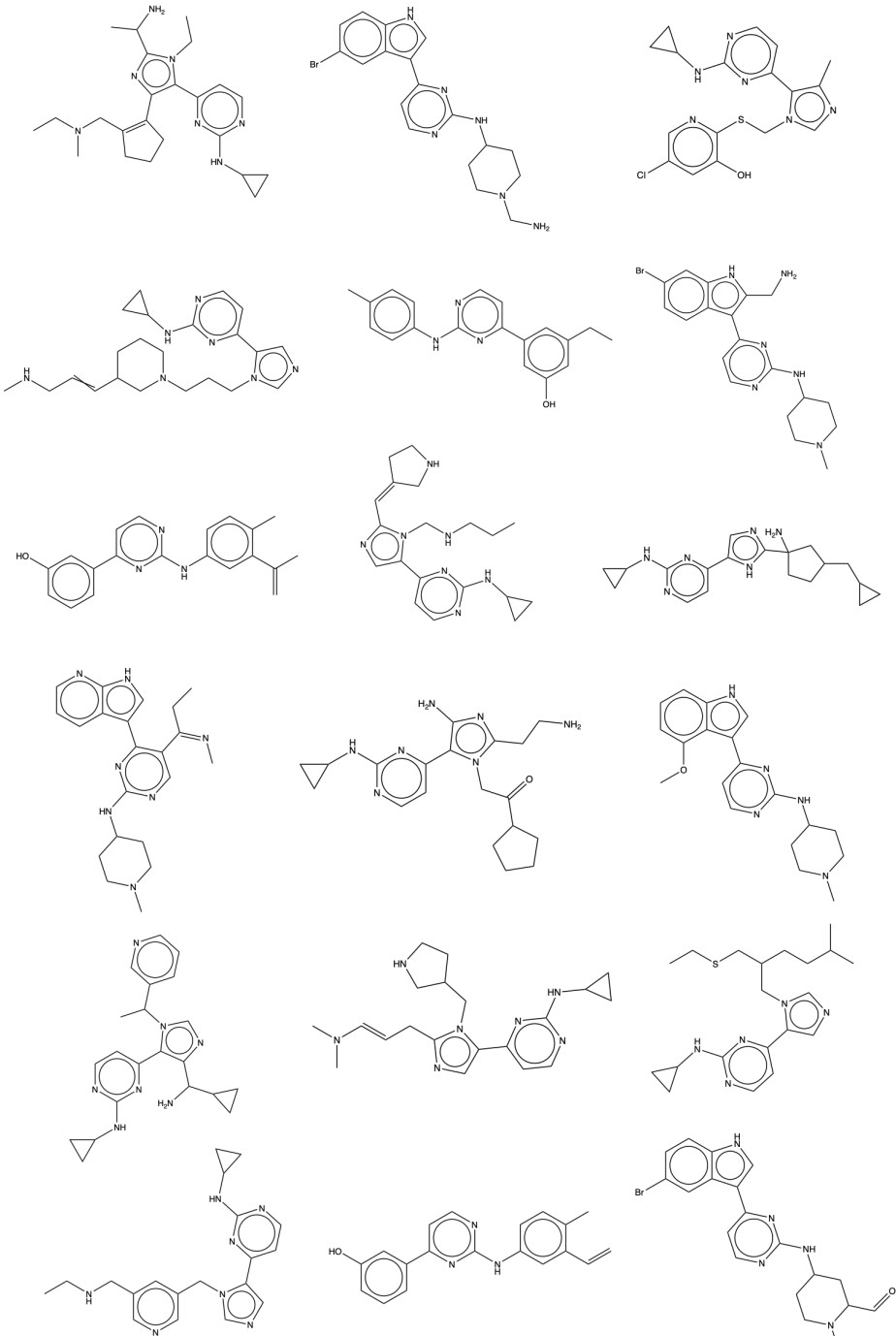

Figure 7: Randomly sampled molecules generated by FaST on the GSK3$\beta$+JNK3+QED+SA task.

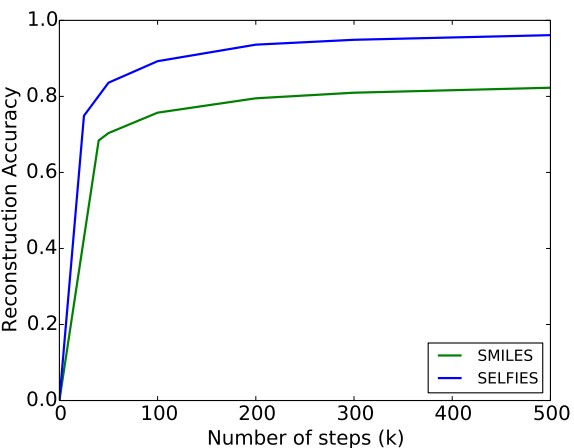

Figure 8: Comparison of reconstruction accuracy when using SMILES string vs SELFIES string. The decoder trained with SELFIES grammar has better reconstruction accuracy (96% vs 86%).

