# OpenReview forum: "Fragment-Based Sequential Translation for Molecular Optimization"
_ICLR.cc/2022/Conference — ICLR 2022 Submitted_

### Official Review · Reviewer_UFDm · 2021-10-26

**Correctness:** 3
**Technical Novelty And Significance:** 3
**Empirical Novelty And Significance:** 4
**Recommendation:** 6
**Confidence:** 4

**Main Review:**

## Strengths

### Fragment generation model
 Fragment-based generation with a fragment generation model is novel and seems to be a reasonable generative model of molecules. It successfully inherits the benefit of the existing fragment-based generative model while avoiding from the limited generative capability due to the fixed fragment dictionary. I also like the authors validate this approach empirically in Section 6.

### Realistic benchmark task
 I like the authors working on a realistic benchmark task, in addition to the traditional penalized logP optimization task.

## Weaknesses
I have concerns on the reinforcement learning module and experiments.

### Non-stationary environment
As the authors state in the "implementation details" paragraph, the environment is non-stationary, i.e., both the initial state distribution and the reward function change during training. This non-stationarity is generally not favorable, and should be avoided if possible, because I don't think RL on non-stationary environments is not well understood. I would avoid the non-stationarity by devising the definition of the environment.

### No theoretical justification
The authors employ the UCB score to define the initial state distribution, but I don't see any theoretical background to justify it.

### Reward design
The reward is designed so that the agent can discover molecules with desirable properties, while being novel and diverse enough. On the other hand, the agent aims to maximize the cumulative reward. I don't see a reasonable interpretation of the cumulative reward, and I would like the authors to discuss it.

### Potentially unfair experimental design?
As the authors state in "optimize for different novelty/diversity metrics" paragraph, the baseline models are not novelty/diversity-aware during training. I am not very sure this is a fair comparison, because the existing RL-based method can be novelty/diversity-aware by introducing the proposed reward design (Eq. 7). In my opinion, the main contribution of this paper is the generative model of fragments, and it would be more valuable (to me) if the experiment is designed so that it highlights the benefit of the generative model of fragments by for example using the same RL component across different methods.

**Summary Of The Paper:**

The present paper aims to discover novel molecules with desirable properties. One of the main contributions is to compose a molecule fragment-by-fragment, where each fragment is drawn not from a fixed fragment dictionary but from a NN-based pre-trained fragment generator. Other contributions are related to the optimization module using RL techniques; introducing fragment deletion as an action, sequential translation, novelty- and similarity-aware reward design, and initial state distribution.
The proposed method is evaluated by ligand generation tasks, where generated molecules are evaluated in terms of success rate, novelty, and diversity.

**Summary Of The Review:**

My recommendation is weak accept.

I like the idea of a generative model of fragments, and the analysis on its benefit in Section 6. However, I am not very convinced by the RL component, because of three concerns raised above. Therefore, I can't aggressively recommend to accept this paper.

My humble suggestion is to set the generative model of fragments as the main contribution and the RL components as minor ones (or even re-using the existing RL component), because the generative model of fragments itself seems to be novel enough (at least to me) to be presented to the community.

---

> ### Author Response · Authors · 2021-11-22
> **Response to Reviewer UFDm**
>
> We thank reviewer UFDm for their thoughtful and constructive feedback. We will address the reviewer's concerns as follows:
>
> **1. On the non-stationary RL environment:**
>
> A non-stationary reward/initial state distribution is common for popular intrinsic reward-based RL exploration strategies such as Inverse-dynamics curiosity [1], random network distillation [2], Go-Explore [3], etc. where the non-stationariness is a natural consequence of using “intrinsic reward” for better exploration. Its effectiveness in hard-exploration problems has been demonstrated in many previous works. We introduce a non-stationary environment to FaST for the same purpose of more efficient exploration in the chemical space to discover diverse and novel compounds. We also updated our paper to reflect the connection to previous work in RL exploration.
>
> [1] Pathak, Deepak, et al. "Curiosity-driven exploration by self-supervised prediction." International conference on machine learning. PMLR, 2017.
>
> [2] Yuri Burda, Harrison Edwards, Amos J. Storkey, Oleg Klimov. “Exploration by random network distillation.” International Conference on Learning Representations. 2019.
>
> [3] Ecoffet, Adrien, et al. "First return, then explore." Nature 590.7847 (2021): 580-586.
>
> **2. On the reward design:**
>
> As we give a binary reward for finding a molecule with desirable properties and novelty/diversity, the expected future cumulative reward (i.e., the value function) can be interpreted as the expected number of molecules that have satisfying properties and novelty/diversity that can be found in the future steps of the current episode.
>
> **3. On using a UCB score for the initial distribution:**
>
> Given the interpretation above about the cumulative reward, the UCB score is an upper confidence bound on the number of good candidates that one can discover by initializing a search trajectory from a given molecule. We keep a finite set of starting molecules that is composed of a given initial set and model-discovered good candidates, while the UCB criterion is used as a heuristic to select starting molecules that give a higher chance of finding more good candidates.
>
> **4. On the experimental design and fair comparison to other methods:**
>
> Thank you for your thoughtful advice, we have added new experiments that compare sample complexity of different methods to (1) create a fair and more informative comparison between methods and (2) demonstrate the effectiveness of the components of FaST. We designed a new evaluation protocol (Figure 2) focusing on sample complexity: we let models keep generating molecules until a set of molecules achieving a certain performance threshold is generated, and count the number of molecules that a model needs to search over. A better-performing model with better sample complexities will search through fewer molecules to gather a good candidate set of molecules. We show that FaST significantly outperforms baseline methods Rationale-RL and MARS in this setting. Note that, in this setting, all methods generate molecules in a diversity/novelty-aware way by filtering out unwanted molecules, and will have the same performance in terms of success rate, diversity, and novelty.
>
> **5. On the presentation of this paper, highlighting distributional fragment vocabulary learning:**
>
> Thank you for your very valuable suggestions. We agree with reviewer UFDm that the vocabulary learning part is the most significant innovation of this work compared to previous methods while being complementary to other advances in fragment-based drug discovery. On the other hand, we also conduct new experiments (Section 6, Figure 2) that better demonstrate how our proposed RL framework can efficiently operate over the learned vocabulary. Specifically, we augment the MARS baseline, which uses MCMC sampling to use both a larger vocabulary and our VQ-VAE learned vocabulary. In both cases, as seen in Figure 2, the model exhibits much worse sample complexity. We have updated the paper to highlight vocabulary learning with VQ-VAE and further clarify the distinction from previous work.

---

> > ### Comment · Reviewer_UFDm · 2021-11-24
> > **Re: Response to Reviewer UFDm**
> >
> > I would appreciate the authors' effort of these additional experiments.
> >
> >
> > I haven't understand why these address my concern.
> > Since the proposed method is trained in a diversity- and novelty-aware way, while the other methods are not, it is almost obvious that the proposed method is better than the others thanks to the training scheme, not to the distributional fragment vocabulary. In fact, the experimental result in Figure 2 shows that the performance difference between FaST and FaST+50k is not much significant as compared to the difference between FaST and MARS families, which suggests that the RL component is more important than the distributional fragment vocabulary. However, the main technical contribution is the distributional fragment vocabulary, rather than the RL component, which is not very consistent with the experimental results.

---

> > > ### Author Response · Authors · 2021-11-27
> > > **Response to Reviewer UFDm's reply**
> > >
> > > Thank you reviewer UFDm for your reply.
> > >
> > > We want to clarify the two main contributions of this work and the different aspects that these contributions are on:
> > >
> > > - Learning of fragment vocabulary. We believe this procedure is significant in its wide applicability to other fragment-based models in molecule-related problems. In this paper, we demonstrate through experiments the usefulness of the learned vocabulary in discovering valid, diverse, and novel molecules under property constraints. Figure 3 and Figure 4 also give intuitive explanations over the benefit of the learned vocabulary, which may not be reflected in the experiment results.
> > >
> > > - While the vocabulary learning component is important in its potential usage in other tasks/applications, we want to clarify it is not trivial to design an RL method that can properly handle novelty/diversity during its optimization, or handle a diverse action space. The main innovations in our RL pipeline include a) a growing frontier from which new searches are initialized from discovered good molecules; b) a UCB algorithm to select a good initial basis; and c) a multi-step policy head to utilize the learned combinatorial vocabulary. Our RL algorithm is designed for the purpose of molecular optimization and is flexible to apply to other molecular optimization problems, such as the constrained penalized logP task (Appendix B).
> > >
> > > In summary, we believe our vocabulary learning method is significant in its potential in developing new and better fragment-based algorithms in molecular applications, while our RL algorithm, which is carefully designed for the task of molecular optimization, is shown to utilize the learned vocabulary well and contribute significantly to our superior performance in experiments.

---

### Official Review · Reviewer_MhuZ · 2021-11-01

**Correctness:** 3
**Technical Novelty And Significance:** 2
**Empirical Novelty And Significance:** 2
**Recommendation:** 3
**Confidence:** 4

**Main Review:**

Combing variational auto-encoders and RL policy for fragment-based molecular optimization gives a good practical performance, but the novelty contribution is marginal. Their method of encoding fragments into latent space is not particularly novel and has been applied in other works for similar tasks (e.g. [1][2][3]). Fragment-based drug design is also a very common approach that has been widely explored for a long while from the cheminformatics community, and there is nothing unique about the fragmentation or the definition of molecular editing action in this work (For example, there are three editing actions defined in [4] and seven in [5] ). The diversity-aware reinforcement learning methods are also common in NLP community [6]. FaST introduces a similarity regularization in the reward function to increase the diversity, which makes a little sense but makes it easy for the model to fool your defined metrics (the difference between AP-based and Morgan-based similarity calculations isn't that great). Such a high level of Nov. and SR. is not proof that your model really works, but rather more like some leakage has occurred, as you did with Table 2 in Appendix B (this experiment actually makes a lot of sense, as the logP task has always been considered less suitable for evaluating molecular optimization).

Minor
1.	It would be better if the authors provide a more intuitive and clear illustration of the motivation of introducing each module in the model (e.g. the VQ-VAE).
2.	Figure 1 is confusing, why does the arrow at the bottom point to a blank space?



[1] Bradshaw, John, et al. "Barking up the right tree: an approach to search over molecule synthesis dags." arXiv preprint arXiv:2012.11522 (2020).

[2] Bradshaw, John, et al. "A model to search for synthesizable molecules." arXiv preprint arXiv:1906.05221 (2019).

[3] Fu, Tianfan, Cao Xiao, and Jimeng Sun. "Core: Automatic molecule optimization using copy & refine strategy." Proceedings of the AAAI Conference on Artificial Intelligence. 2020.

[4] Fu, Tianfan, et al. "MIMOSA: Multi-constraint Molecule Sampling for Molecule Optimization." arXiv preprint arXiv:2010.02318 (2020).

[5] Leguy, Jules, et al. "EvoMol: a flexible and interpretable evolutionary algorithm for unbiased de novo molecular generation." Journal of cheminformatics 12.1 (2020): 1-19.

[6] Shi, Zhan, et al. "Toward diverse text generation with inverse reinforcement learning." arXiv preprint arXiv:1804.11258 (2018).


**Summary Of The Paper:**

This paper proposes to perform molecular optimization by sequential translation. A reinforcement learning policy is employed to increase the diversity of the generated molecules. The proposed method is shown to be able to generate high-quality molecules on single-property and multi-property optimization tasks. The paper is well written and easy to follow.

**Summary Of The Review:**

In general, I do not see too much theoretical or technical contribution. I would request the authors to carefully position their work with the related work mentioned above.

---

> ### Author Response · Authors · 2021-11-22
> **Response to Reviewer MhuZ**
>
> We thank reviewer MhuZ for their thoughtful review and for pointing out a list of very related work. We have added them to our updated manuscript.
>
> The response in summary:
>
> - To our knowledge, all previous fragment-based molecular generative models use a fixed set of fragments, while our proposed method is the first to have a flexible vocabulary that is capable of generating arbitrary fragments and thus able to generate molecules from a much larger chemical space. The vocabulary learning module is complementary to other research in fragment-based drug design and is found natural (eP4U), properly motivated (KpDT), and novel (UFDm) by other reviewers. We thoroughly compare our method to [1,2,3,4,5,6] in this response.
> - We believe the capability to incorporate flexible objectives is a strength of our model, which we further demonstrate through new results and experiments (Figure 2).
> - We give motivation to the proposed method from the common practice of chemists.
>
> Next, we explain these arguments in detail:
>
> **1. Novelty of the proposed method and comparison to previous work**:
>
> Thank you for pointing out the related works to us. A thorough comparison of the techniques used in these methods and our proposed method will certainly improve the clarity of our manuscript. The novelty of FaST lies on:
>
> (1) **Representation learning over a diverse distribution of molecular fragments specially designed for learning a vocabulary for molecular editing and optimization.** Our fragment VQ-VAE encodes and decodes a huge pool of molecular fragments (>50k) in order to construct an embedding space where fragments with similar molecular graphs are embedded as similar vectors. All our fragments also contain an explicit attachment point that is very useful when we compose a base molecule with these fragments together. Moreover, the use of discrete latents (VQ-VAE) gives us more diverse fragment samples and stability in policy learning (we will explain more about this in the rest of this response). Empirical evaluation verifies our method is capable of generating more diverse molecules, and, as a consequence of more efficient exploration in the chemical space, our method is able to find better candidates for various molecular optimization tasks. **To our knowledge, all previous fragment-based molecular generative models use a fixed set of fragments, while our proposed method is the first to have a flexible vocabulary that is capable of generating arbitrary fragments, which reviewer UFDm found “novel enough to be presented to the community”.**
>
> As a comparison to previous work that embed molecular fragments / does fragment-based drug design:
>
> [1,2] design autoencoders for a molecular synthesis DAG or a single-step chemical reaction, where the encodings of the reactants are combined with other components to facilitate the representation learning for DAGs / single-step reactions. These two methods use a fixed set of reactants, and decoding is done through selection from this fixed set. When decoding a DAG / single-step reaction, the molecular reactants contained in it are decoded in a way that predicts the logit for selecting a reactant from a fixed predefined set of reactants.
>
> [3] encode and decode entire molecules instead of fragments while using fragments (substructures) as a sub-procedure in the decoding process. The fragments themselves are never decoded. The set of available substructures is finite and predefined. According to Table 1 of [3], there are less than 1000 substructures for all experiments in [3]. [4,5] are MCMC samplers that do fragment-based editing to optimize molecules. But again they select fragments from a fixed set for the editing. On the contrary, the learned policy of FaST predicts a fragment embedding vector, which the fragment autoencoder decodes into a molecular fragment. Our fragment vocabulary is very flexible and thus our generation is not limited by a fixed vocabulary.
>
> **Fragment-based drug design is an existing approach. However, The use of a fixed set of fragment vocabulary is a long-standing limitation of fragment-based molecular generative models, which we resolve with representation learning using a VQ-VAE.** We see our method for learning fragment vocabulary as complementary to previous work that uses a fixed finite fragment vocabulary. We also believe the various editing actions, such as the 7 actions in [6] as complementary to our method.
>
> (2) Another technical contribution of this paper is a RL algorithm that operates over the learned fragment vocabulary and can optimize various user objectives, including diversity/novelty. **Diversity-driven RL is widely studied in the RL exploration context [7,8,9,10] but underexplored in the molecular generation context.** It is also due to the flexibility of our learned vocabulary that FaST is capable of generating very diverse compounds. We believe this capability leads to the superior performance of FaST demonstrated in our experiments.

---

> > ### Author Response · Authors · 2021-11-22
> > **Response to Reviewer MhuZ continued**
> >
> > **2. On the concern about optimization exploiting the evaluation metrics (leakage):**
> >
> > On one hand, Evaluation for molecular generative models is known to be a hard task. Most current benchmarks rely on ML property predictor-based or fingerprint-based metrics for which exploitation of the metric is a fair concern. **More recent evaluation protocols, such as the multi-objective optimization task [11] explored in this paper are more robust than single-objective optimization due to the extra regularization power of more constraints.**
> >
> > On the other hand, we believe the capability to optimize for diversity or other objectives is a strength of our proposed method. Baseline methods such as RationaleRL and MARS achieve decent novelty/diversity under Morgan fingerprints but fail to achieve good performance under the AP fingerprint while being diversity-unaware. **We additionally report new results in Table 3, where FaST optimizing for Morgan-novelty has poor performance when evaluated over AP-novelty, but FaST trained to optimize for AP-novelty still performs well when evaluated over Morgan-novelty.** We believe this result demonstrates that AP-novelty is an objective that is harder for the model to trick and it is non-trivial to achieve this stricter diversity/novelty metric. Such a strict objective necessitates a more flexible generative model and diversity-awareness, which leads to the larger performance gap between FaST and competing methods. We believe being able to optimize for the more robust AP-novelty demonstrates the importance of FaST's capability to incorporate arbitrary constraints through reward design.
> >
> > **3. Motivation of modules of FaST:**
> >
> > **Vocabulary**: Using a substructure-defined vocabulary makes more sense than atom-by-atom generation because chemists think on the level of functional groups/substructures instead of single atoms. We want to take this idea one step further compared to previous methods because fixed vocabulary limits the editing capabilities of the model. In order to learn a flexible vocabulary that is easy for further usage in optimization, we make use of VAE, and in particular VQ-VAE because the discrete latent structure is suitable for encoding the molecular fragments, which are discrete objects. The discrete latent also enables the RL policy to optimize through discrete actions, which is more stable in training than operating over continuous actions.
> >
> > **Sequential translation (our RL search strategy)**: Chemists do not design new molecules from scratch each time, but often start from some molecules they have some prior knowledge about. We mimic this process by always sampling from a distribution of good molecules that we have already found with our search algorithm so that we never generate from scratch but by utilizing prior knowledge. This is implemented through our expanding “frontier” which stores good molecules discovered by the model, and sampling from the frontier is guided by the UCB score.
> >
> > **4. Regarding Figure 1:**
> >
> > Sorry for the confusion and thank you for your feedback. In our proposed method new molecules discovered by the model are added to the frontier, where new search trajectories are initialized from. The arrow should point to the frontier at the leftmost of Figure 1. We have updated Figure 1 to clarify this point.
> >
> > References:
> >
> > [1] Bradshaw, John, et al. "Barking up the right tree: an approach to search over molecule synthesis dags." arXiv preprint arXiv:2012.11522 (2020).
> >
> > [2] Bradshaw, John, et al. "A model to search for synthesizable molecules." arXiv preprint arXiv:1906.05221 (2019).
> >
> > [3] Fu, Tianfan, Cao Xiao, and Jimeng Sun. "Core: Automatic molecule optimization using copy & refine strategy." Proceedings of the AAAI Conference on Artificial Intelligence. 2020.
> >
> > [4] Fu, Tianfan, et al. "MIMOSA: Multi-constraint Molecule Sampling for Molecule Optimization." arXiv preprint arXiv:2010.02318 (2020).
> >
> > [5] Xie, Yutong, et al. "Mars: Markov molecular sampling for multi-objective drug discovery." arXiv preprint arXiv:2103.10432 (2021).
> >
> > [6] Leguy, Jules, et al. "EvoMol: a flexible and interpretable evolutionary algorithm for unbiased de novo molecular generation." Journal of cheminformatics 12.1 (2020): 1-19.
> >
> > [7] Shi, Zhan, et al. "Toward diverse text generation with inverse reinforcement learning." arXiv preprint arXiv:1804.11258 (2018).
> >
> > [8] Pathak, Deepak, et al. "Curiosity-driven exploration by self-supervised prediction." International conference on machine learning. PMLR, 2017.
> >
> > [9] Yuri Burda, Harrison Edwards, Amos J. Storkey, Oleg Klimov. “Exploration by random network distillation.” International Conference on Learning Representations. 2019.
> >
> > [10] Ecoffet, Adrien, et al. "First return, then explore." Nature 590.7847 (2021): 580-586.
> >
> > [11] Jin, Wengong, Regina Barzilay, and Tommi Jaakkola. "Multi-objective molecule generation using interpretable substructures." International Conference on Machine Learning. PMLR, 2020.

---

### Official Review · Reviewer_KpDT · 2021-11-02

**Correctness:** 3
**Technical Novelty And Significance:** 2
**Empirical Novelty And Significance:** 3
**Recommendation:** 6
**Confidence:** 4

**Main Review:**

Strength:
1. The proposed model outperforms other baselines in the multi-objective molecules optimization benchmark.
2. Applying VQ-VAE into molecular fragment vocab learning is a properly motivated hypothesis and empirically well-validated.
3. MolDQN[1] and CMG[2] are other multi-objective optimization models. Better to include them at least in the related work.

[1] Zhou, Zhenpeng, et al. "Optimization of molecules via deep reinforcement learning." Scientific Reports 9.1 (2019): 1-10.
[2] Shin, Bonggun, et al. "Controlled molecule generator for optimizing multiple chemical properties." Proceedings of the Conference on Health, Inference, and Learning. 2021.


Weakness:
1. In the Encoder/Decoder section on page 4, the authors claimed as "we ﬁnd the SELFIES decoder works well empirically". It's not convincing why they choose to use the SELFIES decoder. It'd be good if it's well-motivated and empirically shown.
2. Minor typo: In eq (6), the outer parenthesis of the diversity may be before the "<" sign?
3. The overall framework is novel, however, it relies on many known methods.

**Summary Of The Paper:**

This paper proposes a new reinforcement learning method for molecular optimization. They train VQ-VAE to learn the vocabulary of molecular fragments which enables the optimizer to efficiently search the chemical space. The optimizer consisted of general reinforcement learning and MPNN, and devised sampling schemes to improve performance. By integrating evaluation criteria as a reward function, the proposed model can additionally improve the performance in terms of those criteria. The experiments show not only its superior performance but shorter optimization time.


**Summary Of The Review:**

 Overall it is a well-written paper with good motivation, appropriate method, and supportive experiments. The overall framework is novel, however, it relies on many existing methods.

---

> ### Author Response · Authors · 2021-11-22
> **Response to Reviewer KpDT**
>
> We thank reviewer KpDT for their useful feedback. We will address the reviewer's concerns as follows:
>
> **1. MolDQN and CMG**:
>
> Thank you for pointing out these related works, we have added them to the updated manuscript and added MolDQN to the result comparison in Table 2, for the constrained penalized logP task.
>
> **2. Motivation for a SELFIES decoder and comparison to other decoders**:
>
> We expand on our architecture choice more clearly in appendix F in the updated manuscript: Our SELFIES decoders achieve a higher reconstruction accuracy than SMILES decoders (Figure 8). A simple architecture fits our purpose of generating small fragments (<= 10 atoms) well. Graph decoders are generally very complex, often requiring careful design over the grammar, handling of edge cases, and their own generation vocabulary. We use SELFIES strings over SMILES, because the grammar of SELFIES strings is more robust in that it is less likely for sampled strings from the SELFIES decoder to be invalid.
>
> **3. Typos and writing**:
>
> Thank you for pointing out the typo in equation (6). We have fixed this typo and restructured the writing to better clarify the novelty of this work and the connection to previous work.

---

### Official Review · Reviewer_eP4U · 2021-11-03

**Correctness:** 3
**Technical Novelty And Significance:** 2
**Empirical Novelty And Significance:** Not applicable
**Recommendation:** 3
**Confidence:** 4

**Main Review:**

Pros:
- VQ-VAE is a natural model for encoding discrete objects and produces more diverse structures, which is confirmed by t-SNE plots.
- The authors use Upper Confidence Bound (UCB) to sample compounds effectively and accelerate the search of compounds with optimized properties.
- There is an interesting discussion on the usability of logP optimization and the flaws of measuring compound similarity using molecular fingerprints.

Cons:
- The text structure is very similar to the MARS paper, and there is an apparent similarity between these two methods, although there are some significant changes.
- One of the similarities is the main results table, which includes metrics that do not correctly emphasize the superiority of one method above others. The success rate (SR), which should be the main objective, is equal (almost) 1 for many models, so it is difficult to say if one method is significantly better than others.
- Besides SR, other metrics describe the novelty and diversity of the generated compounds. These metrics are directly optimized by the proposed model. Are other methods in the comparison trained so that these metrics are maximized? If not, then it is not surprising that FaST wins in this competition. Also, the last metric, PM, is the product of all the other metrics, so suboptimal diversity or novelty scores of other models negatively impact the scoring.
- To confirm the usefulness of the model, a different comparison should be proposed, or more biological targets should be added. One idea would be to create a benchmark that uses uncertainty prediction models. This way, the novel compounds that cannot be reliably predicted by the activity models would be discarded.
- In Section 4, some claims are not supported experimentally in the paper:
  - "we find that using a VQ-VAE with a categorical prior instead of the typical Gaussian prior makes RL training stable and provides good performance gains" - the FaST variant with the Gaussian prior should be included in the results table,
  - "we find the SELFIES decoder works well empirically" - this could be further discussed in the ablation studies.
- More examples of generated compounds would be welcome.

Other comments:
- In the paragraph about VQ-VAE in Section 3, should it not be $k^d$ instead of $d^k$ distinct fragments?
- typos: "specify the the molecule to keep"

**Summary Of The Paper:**

The paper presents a new way of molecular optimization that is based on chemical fragments. The authors employ VQ-VAE to encode a fragment library in a coherent latent space. They argue that this method produces more diverse rationales than a Gaussian VAE. The proposed generative method uses reinforcement learning (PPO) to select modifications of the chemical structure: a fragment decoded by VQ-VAE is added at a chosen attachment point, or a part of the compound is removed. The model retains only modifications that fulfill initial constraints regarding the novelty, diversity, and chemical properties of a generated molecule measured by formerly trained predictive models. The experiments show significantly better performance of the proposed FaST model compared to a diverse set of baseline models.

**Summary Of The Review:**

Based on the above comments, I currently recommend rejecting this paper. The novelty of the work lies in incremental changes to existing methods, and I think the modifications are substantial. However, the experiments, in my opinion, do not demonstrate well the impact of these changes.

---

> ### Author Response · Authors · 2021-11-22
> **Response to Reviewer eP4U**
>
> We thank reviewer eP4U for their constructive comments and suggestions. We first highlight responses to several main concerns and then provide point-to-point responses.
>
> This response in summary:
>
> - We clarify the evaluation metrics and the importance of novelty/diversity in molecular optimization. We introduce a new evaluation protocol to compare methods on their sample complexity given a performance threshold, under which all methods are novelty/diversity-aware and would achieve the same SR, Div and Nov. FaST still outperforms baseline methods significantly in sample complexity under this setting.
> - We add more ablation where we augment a baseline method with our flexible vocabulary and our method with a fixed vocabulary and demonstrate that the combination of the techniques constitutes superior performance.
> - We address other concerns such as VQ-VAE vs normal VAE (Section 4.1), SELFIES decoder vs other decoders (Appendix F, Figure 8) and include more samples in the appendix (Figure 7).
>
> Next, we address the reviewer’s concern in detail:
>
> **1. Similarity to MARS and what makes FaST different:**
>
> FaST is similar to MARS as both methods learn policies under a similar action space: either adding a fragment to the current molecule or deleting a fragment through deleting a bond. On the other hand, we believe the novel components of distributional fragment vocabulary and RL policy distinguish FaST from MARS:
>
> FaST learns a distributional fragment vocabulary and synthesizes novel molecules with this fragment distribution, which enables the generation of much more diverse molecules. On the other hand, MARS operates over a fixed set of 1000 molecular fragments.
> MARS is a MCMC sampler where a value function is not learned and thus is unable to count for future rewards. When the molecular optimization task becomes harder, it may become increasingly difficult to find optimal candidates through single-step feedback only. In our new ablations, we added experiments (Figure 2/Section 6) where we use larger vocabulary with MARS (a larger fixed vocabulary and our VQ-VAE distributional vocabulary), and find that MARS does not perform well under either of these two settings. Therefore, FaST’s RL strategy searches for good molecules significantly more efficiently.
>
> **2. The success rate (SR), which should be the main objective, is equal (almost) 1 for many models, so it is difficult to say if one method is significantly better than others:**
>
> We adopted the multi-objective molecular optimization from previous work [1] and we did not modify the task definition for a fair comparison to other methods. While success rate is very important, novelty and diversity are also very important for the task of drug discovery. These metrics were designed to emphasize novelty and diversity because only evaluating success rate will encourage the generation of uninteresting molecules, For instance, if we add a single carbon to a molecule, the output of the oracle property predictor is unlikely to deviate a lot (especially for a pre-trained ML property predictor). Therefore, the model will simply generate a set of molecules that are all too similar to each other, and the generated molecules would be unhelpful for de novo drug design.
>
> Furthermore, since many methods are able to find good candidates for this series of tasks, we believe our added experiments (Figure 2) evaluating sample complexity give a more informative comparison of the methods. If a scoring function is available, one can find good candidates by enumerating all possible chemical compounds while filtering out bad ones. A smart search algorithm will be able to find good candidates much faster than a random search. Under this setting, we find FaST is able to discover successful and diverse molecules with many fewer samples than baseline methods.
>
> **3. novelty and diversity are directly optimized by the proposed model while not for other models in the comparison. To confirm the usefulness of the model, a different comparison should be proposed, or more biological targets should be added:**
>
> Thank you for your constructive suggestions, We add more experiments (Figure 2) that augment previous methods with components proposed in FaST and a new evaluation looking at the sample complexity of different models and ablations. Under this evaluation protocol, all methods keep proposing molecules until a good enough set is found and are diversity/novelty-aware. All methods will have the same success rate, diversity, and novelty, while a better method will need fewer samples to gather the desired set of molecules. We observe that under this more fair setting comparing the different methods that our method, FaST is still able to outperform the baselines.

---

> > ### Author Response · Authors · 2021-11-22
> > **Response to Reviewer eP4U continued**
> >
> > **4. The FaST variant with the Gaussian prior:**
> >
> > We experimented with FaST operating over a VAE trained with Gaussian prior, in which case the RL policy will need to output a Normal distribution from which the action is sampled. In our experiments, the training instability under this setting led to training failure. Note that the PPO objective [1] is:
> >
> > $$L(\theta) = \hat{\mathbb{E}}[ \min( \frac{\pi_\theta(a_t | s_t)}{\pi_o(a_t | s_t)} \hat{A}_t,
> > \mathrm{clip} (\frac{\pi_\theta(a_t | s_t)}{\pi_o(a_t | s_t)}, 1-\epsilon, 1+\epsilon)
> > \hat{A}_t)]$$
> >
> > Where $\pi_o$ is the old policy before the policy update. We notice that the likelihood ratio can easily explode for FaST under Gaussian VAE and continuous action space. We did not include this Gaussian variant in the result table because we were not able to get meaningful results due to the training instability. We explain this in more detail in the updated draft (Section 4.1).
> >
> > **5. SELFIES decoder vs other fragment decoders:**
> >
> > We expand on our architecture choice more clearly in appendix F in the updated manuscript: Our SELFIES decoders achieve a higher reconstruction accuracy than SMILES decoders (Figure 8). A simple architecture fits our purpose of generating small fragments (<= 10 atoms) well. Graph decoders are generally very complex, often requiring careful design over the grammar, handling of edge cases, and their own generation vocabulary. We use SELFIES strings over SMILES because the grammar of SELFIES strings is more robust in that it is less likely for sampled strings from the SELFIES decoder to be invalid.
> >
> > **6. More samples from FaST, typos:**
> >
> > Thank you for your suggestions. We include more randomly sampled molecules generated from FaST for the GSK+JNK+QED+SA task in the appendix (Figure 7). We also fix the typos.
> >
> > **References:**
> >
> > [1] Jin, Wengong, Regina Barzilay, and Tommi Jaakkola. "Multi-objective molecule generation using interpretable substructures." International Conference on Machine Learning. PMLR, 2020.
> >
> > [2] Schulman, John, et al. "Proximal policy optimization algorithms." arXiv preprint arXiv:1707.06347 (2017).

---

> > > ### Comment · Reviewer_eP4U · 2021-11-28
> > > **Reply**
> > >
> > > Thank you for your reply and for updating the manuscript. The motivation is now much clearer, and there is evidence supporting some statements that were indicated by me and other reviewers. However, I am still not convinced by the experimental section. The additional experiment shown in Figure 2 is the first step to prove the usefulness of the proposed approach, but more such evaluations focusing on the built fragment library would be appreciated.  I am more positive about the method now, especially after seeing some of the generated compounds, but I do not feel this is enough to change my score.

---

### Author Response · Authors · 2021-11-22
**General Comments to all reviewers:**

We thank all reviewers for their thoughtful and constructive comments. We have made the following major revisions to the manuscript to address the reviewers’ concerns:

- To address the concern on evaluation metrics, we designed a new evaluation protocol (Figure 2, Section 5) where we let models keep generating molecules until a set of molecules achieving a certain performance threshold is generated, and count the number of molecules that a model needs to search over. A better-performing model with better sample complexities will search through fewer molecules to gather a good candidate set of molecules. We show that FaST (need 71k searches) significantly outperforms baseline methods RationaleRL (205k searches) and MARS (759k searches) in this setting. Note that, in this setting, all methods generate molecules in a diversity/novelty-aware way by filtering out unwanted molecules, so this evaluation more fairly compares every model.

- We revise the related work section (Section 2) to include relevant literature pointed out by the reviewers, and clarify the main novelty of this paper: we learn a flexible fragment vocabulary using a VQ-VAE and use it with RL for editing molecules, compared to previous fragment-based molecular optimization methods that all use a fixed fragment vocabulary, which limits the generation capability.

- To further validate the effectiveness of the novel changes introduced in this paper, we added more ablation studies (Figure 2, Section 6) where we augment baseline methods with our learned vocabulary, as well as our method with a fixed vocabulary, in order to demonstrate the importance of both the distributional vocabulary and our RL search algorithm. We find that all components of our proposed method are crucial to our superior performance.

We also made other revisions throughout our manuscript and appendices to address other comments by the reviewers. New changes are highlighted in red.

---

### Decision · Program_Chairs · 2022-01-20

**Decision:**

Reject

**Comment:**

Reviewers agree that the paper is well-motivated and the proposed method is somewhat interesting and well-experimented. However, reviewers feel that the paper relies on many existing methods and does not appear to be novel enough.